# Structure and elasticity of CaC$_2$O$_5$ suggests carbonate contribution to the seismic anomalies of Earth's mantle

Hanyu Wang [1,2], Lei Liu [2] ✉, Zihan Gao[2], Longxing Yang[1,2], Gerile Naren[2] & Shide Mao [1] ✉

Knowledge of carbonate compounds under high pressure inside Earth is key to understanding the internal structure of the Earth, the deep carbon cycle and major geological events. Here we use first-principles simulations to calculate the structure and elasticity of CaC$_2$O$_5$-minerals with different symmetries under high pressure. Our calculations show that CaC$_2$O$_5$-minerals represent a group of low-density low-seismic-wave velocity mantle minerals. Changes in seismic wave velocity caused by the phase transformation of CaC$_2$O$_5$-*Cc* to CaC$_2$O$_5$-*I$\bar{4}$2d* (CaC$_2$O$_5$-*C2-l*) agree with wave velocity discontinuity at a depth of 660 km in the mantle transition zone. Moreover, when CaC$_2$O$_5$-*Fdd*2 transforms into CaC$_2$O$_5$-*C2* under 70 GPa, its shear wave velocity decreases by 7.4%, and its density increases by 5.8%, which is consistent with the characteristics of large low-shear-velocity provinces (LLSVPs). Furthermore, the shear wave velocity of CaC$_2$O$_5$-*I$\bar{4}$2d* is very similar to that of cubic Ca-perovskite, which is one of the main constituents of the previously detected LLSVPs. Therefore, we propose that CaC$_2$O$_5$ and its high-pressure polymorphs may be a main component of LLSVPs.

Understanding the physical and chemical characteristics of minerals under high pressure is crucial to understanding the Earth's composition, structure, and dynamic processes. Exploring the Earth's mantle velocity structure relies on quantitative knowledge of the elastic properties of mantle minerals[1].

Through the carbon cycle, carbon is continuously exchanged from the Earth's surface to its interior. Studying the physical properties of carbon-bearing materials such as carbonates under high pressure and high temperature is crucial for gaining insight into the Earth's deep carbon cycle; however, knowledge of the forms, transition mechanisms, and movements of carbon in the deep Earth is still limited[2]. CaC$_2$O$_5$ is a group of polymorphic carbonate minerals first discovered via first-principles simulation, and the stable structure of CaC$_2$O$_5$ with symmetries of *Fdd*2, *Pc*, and *C*2 was established[3]. Yao et al. proposed that CaC$_2$O$_5$ undergoes successive structural phase transformations

with increasing pressure: *Pc* to *Fdd*2 at 38 GPa, *Fdd*2 to *Pc* at 72 GPa, and *Pc* to *C*2 at 82 GPa. Recently, two new polymorphs of CaC$_2$O$_5$ with *Fd$\bar{3}$m* and *I$\bar{4}$2d* symmetries were discovered by high-pressure DAC experiments and first-principles simulations, and researchers believe that CaC$_2$O$_5$-*I$\bar{4}$2d* can stably exist in the lower mantle from 34 to 45 GPa and may promote the carbon cycle and material transformation in the deep mantle[4]. Later, Sagatova et al. proposed[5] two new polymorphs of CaC$_2$O$_5$ with space groups *Fdd*2-*l* and *Cc* at 0 and 15 GPa by first-principles simulation, respectively, and suggested that CaC$_2$O$_5$-*Cc* may exist in the upper mantle and mantle transition zone; at pressures of 25 and 50 GPa, they proposed a new polymorph with symmetry of CaC$_2$O$_5$-*C2-l* and confirmed the structural stability of CaC$_2$O$_5$-*I$\bar{4}$2d*. Although the crystal structure, partial electronic properties, and possible phase transforms of these CaC$_2$O$_5$ polymorphs have been extensively studied, there are no unified results depicting the possible

[1]State Key Laboratory of Geological Processes and Mineral Resources, and School of Earth Sciences and Resources, China University of Geosciences, 100083 Beijing, China. [2]United Laboratory of High-Pressure Physics and Earthquake Science, Institute of Earthquake Forecasting, China Earthquake Administration, 100036 Beijing, China. ✉e-mail: liulei@ief.ac.cn; maoshide@cugb.edu.cn

impact of $CaC_2O_5$ polymorphs on the mantle or their structural stability and elastic properties.

In this study, to understand the possible model of phase transformation of those $CaC_2O_5$ polymorphs and its impact on the mantle structure, composition and deep carbon cycling, the lattice parameters, electronic properties, and elasticity of the 6 polymorphs of $CaC_2O_5$ (hereafter referred to as $CaC_2O_5$s), including $CaC_2O_5$-$Cc$, $CaC_2O_5$-$Fdd2$, $CaC_2O_5$-$C2$-$l$, $CaC_2O_5$-$C2$, $CaC_2O_5$-$I\bar{4}2d$ and $CaC_2O_5$-$Pc$, were calculated by first-principles simulation for their respective stable pressure ranges. First-principles calculations have been successfully applied to geosciences to understand mineral properties such as structural, elastic properties, electronic properties, etc., under high pressure and temperature[6–11]. Our results provide insights into the presence of various structural phases of $CaC_2O_5$ in the mantle and their effects on the deep mantle.

## Results and discussion

### Structural stability of $CaC_2O_5$-$I\bar{4}2d$ under high pressure

$CaC_2O_5$-$I\bar{4}2d$ was first synthesized in the laboratory under two pressures of 34 GPa and 45 GPa[4]. However, Sagatova et al. proposed[5] that this mineral can be stable at 25 to 50 GPa from the first-principles simulations, so its stability range is unclear. The thermodynamic properties of minerals are usually evaluated by analyzing their phonon frequencies across the Brillouin Zone[12–14]. Therefore, the phonon dispersion along selected high-symmetry points in the Brillouin zone of $CaC_2O_5$-$I\bar{4}2d$ was calculated at 30 to 100 GPa (see Supplementary Fig. 1). Our calculated results show that lattice vibrations produce negative values in the Brillouin region under pressures between 0 and 34 GPa, which indicates that $CaC_2O_5$-$I\bar{4}2d$ is unstable in this pressure range[15]. With increasing pressure, the structure shows thermodynamic stability between 34 and 100 GPa.

To further verify the stability of $CaC_2O_5$-$I\bar{4}2d$, we calculated the Mulliken population to explore whether the electronic properties of $CaC_2O_5$-$I\bar{4}2d$ undergo mutations at high pressure (see Supplementary Table 1). The band spilling parameter for spin component 1 was 0.69%, indicating that our calculated results are reasonable and reliable[16].

The Mulliken population analysis of $CaC_2O_5$-$I\bar{4}2d$ at pressures of 34 GPa, 45 GPa, 70 GPa, 85 GPa, and 100 GPa showed that the population of all three C-O bonds was greater than 0.5, with a maximum difference of only 5.17% at different pressures. This result indicates that these bonds are covalent. However, the population values of all three Ca-O bonds are less than 0.1. The population of Ca-O$^a$ and Ca-O$^b$ bonds changes significantly with pressure, while the population of Ca-O$^c$ bonds remains relatively unaffected by pressure. This result indicates that these bonds are ionic. Furthermore, there was no abrupt change in the atomic electronegativity or bond denstiy from 34 to 100 GPa, indicating the stability of $CaC_2O_5$-$I\bar{4}2d$.

In summary, the phonon dispersion and Mulliken population results show that $CaC_2O_5$-$I\bar{4}2d$ maintains structural stability under 34–100 GPa.

### Lattice parameters and density

The lattice parameters ($a$, $b$, and $c$) of $CaC_2O_5$s under high pressure were calculated (see Supplementary Fig. 2). To elucidate the differences in the lattice parameters of $CaC_2O_5$, $2 \times 2 \times 1$ supercells of $CaC_2O_5$-$C2$ and $CaC_2O_5$-$Pc$ and a $2 \times 1 \times 1$ supercell of $CaC_2O_5$-$Cc$ were built to ensure consistent atomic numbers in $CaC_2O_5$ crystal cells. The lattice constants of $CaC_2O_5$s linearly decrease with increasing pressure ($R^2 > 0.978$). Notably, under the same pressure, the lattice parameters, $a$ and $b$, of $CaC_2O_5$-$I\bar{4}2d$ and $CaC_2O_5$-$C2$-$l$ are very close in value, and the ratio of the lattice parameter, $c$, of the two polymorphs is constant at 0.8 (±0.05).

The densities of all 6 calculated $CaC_2O_5$ polymorphs under high pressure are listed in Fig. 1. Moreover, the densities of the main mineral phases of the lower mantle, such as $CaSiO_3$ perovskite[17], $MgSiO_3$

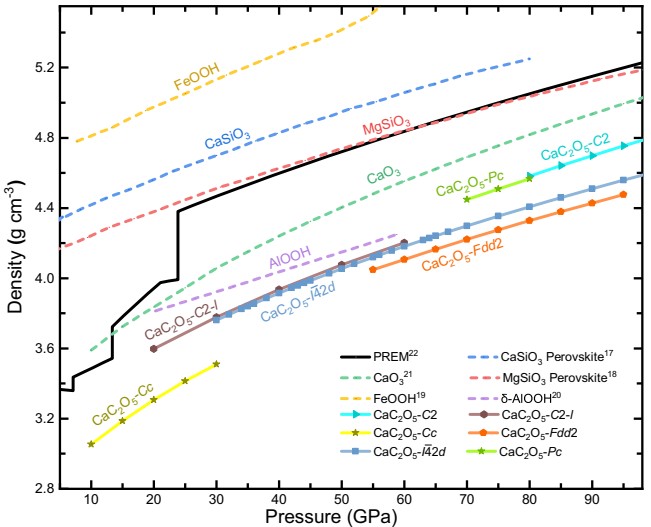

**Fig. 1 | Density of $CaC_2O_5$s and several lower mantle minerals under high pressure.** Comparison of the densitites of $CaSiO_3$ perovskite[17] (bule dash line), $MgSiO_3$ perovskite[18] (red dash line), FeOOH[19] (gold dash line), δ-AlOOH[20] (lavender dash line), $CaO_3$[21] (green dash line) and $CaC_2O_5$s (soild line) with that of Earth's mantle according to the PREM (Black soild line).

perovskite[18], FeOOH[19], δ-AlOOH[20], and the newly discovered $CaO_3$[21] under high pressure, and the density of the Preliminary Reference Earth Model (PREM)[22] are also listed in Fig. 1.

The densities of $CaC_2O_5$s increase with increasing pressure, but all the densities are lower than those of the PREM model. Generally, low-density minerals have difficulty entering the deep mantle. However, several low-density minerals, such as δ-AlOOH[20] and $CaO_3$[21], have also been discovered in the lower mantle. Therefore, due to FeOOH[18] and silicate minerals with densities higher than those of the PREM are widely present in the mantle, $CaC_2O_5$s may play an essential role in regulating and neutralizing the mantle density in the Earth's lower mantle.

As pressure increases, some clear density transitions occur between different polymorphs of $CaC_2O_5$. First, there is a density increase of 8.81%–7.37% from 20 GPa to 30 GPa when $CaC_2O_5$-$Cc$ transforms into $CaC_2O_5$-$C2$-$l$ at approximately 660 km, which is very consistent with the increase in density at 660 km in the PREM model, indicating that this structural phase transformation may be one of the reasons for the changes in density here. Second, when $CaC_2O_5$-$I\bar{4}2d$ or $CaC_2O_5$-$Fdd2$ transforms into $CaC_2O_5$-$Pc$ and $CaC_2O_5$-$C2$, the density increases by 4.02%–5.88%. The densities of $CaC_2O_5$-$C2$-$l$ and $CaC_2O_5$-$I\bar{4}2d$ show very similar quantities and relationships with pressure.

### Elasticity

The elastic parameters of minerals and their dependence on pressure are essential in Earth science for understanding processes ranging from brittle failure to flexure to the propagation of elastic waves. Seismic observations reveal the structure of the Earth, including the radial (one-dimensional) profile, lateral heterogeneity, and anisotropy, which are primarily determined by the elastic parameters of minerals and their dependence on pressure and temperature[23]. Therefore, to understand the geological properties of $CaC_2O_5$s, we calculated the elastic constants of $CaC_2O_5$s under high pressures, as shown in Fig. 2.

Based on a theorem for determining the elastic stability of minerals[24], the elastic stability of $CaC_2O_5$s was investigated, and the results showed that the 6 polymorphs maintain elastic stability within their calculated pressure ranges. Although the space groups of $CaC_2O_5$-$C2$-$l$ and $CaC_2O_5$-$I\bar{4}2d$ are different, their elastic constants show good consistency. Within the same pressure range, $C_{11}$ of

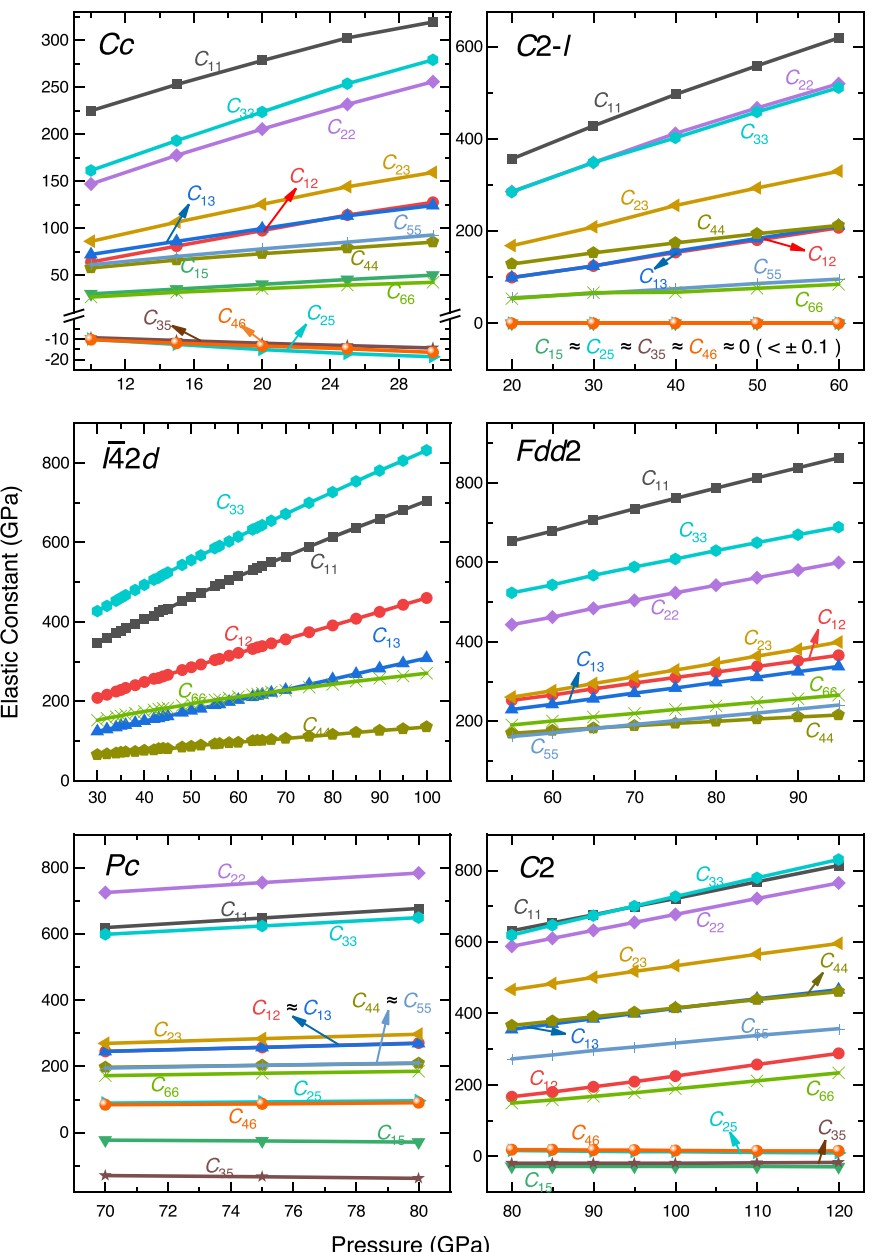

**Fig. 2 | Elastic constants of CaC$_2$O$_5$s.** The elastic constants ($C_{11}$, $C_{12}$, $C_{13}$, $C_{15}$, $C_{22}$, $C_{23}$, $C_{25}$, $C_{33}$, $C_{35}$, $C_{44}$, $C_{46}$ and $C_{66}$) of CaC$_2$O$_5$ at their respective structural phase transition pressures[5] were compared (the space group symbol is shown in the figure). *Pc*, *C2*, *Cc*, and *C2-l* belong to the monoclinic system, with 13 independent elastic constants; *Fdd*2 belongs to the orthorhombic system, with 9 independent elastic constants; *I$\bar{4}$2d* belongs to the tetragonal system, with 6 independent elastic constants. Elastic constants in the same direction are represented by the same color.

*C2-l* ≈ $C_{33}$ of *I$\bar{4}$2d*, $C_{22}$ and $C_{33}$ of *C2-l* ≈ $C_{11}$ of *I$\bar{4}$2d*, $C_{23}$ of *C2-l* ≈ $C_{12}$ of *I$\bar{4}$2d*, $C_{44}$ of *C2-l* ≈ $C_{66}$ of *I$\bar{4}$2d*, $C_{12}$ and $C_{13}$ of *C2-l* ≈ $C_{13}$ of *I$\bar{4}$2d*, and $C_{55}$ and $C_{66}$ of *C2-l* ≈ $C_{44}$ of *I$\bar{4}$2d*; additionally, $C_{15}$, $C_{25}$, $C_{35}$ and $C_{46}$ of *C2-l* are very close to 0.

The elastic modulus is an important parameter for describing the physical and chemical properties of minerals. In crystalline systems, assuming that the arrangement direction is random, the bulk modulus (*K*) and shear modulus (*G*) can be obtained by the Voigt, Reuss, and Hill formulas[25], and the Hill modulus is used here[26] based on the average of the Voight and Reuss moduli. The bulk modulus and shear modulus of CaC$_2$O$_5$s were calculated and are shown in Fig. 3a, b. The *K* and *G* of CaC$_2$O$_5$s linearly increase with pressure. Among the 6 polymorphs, the *K* and *G* of CaC$_2$O$_5$-*Fdd*2 are the largest, and those of CaC$_2$O$_5$-*Cc* are the smallest. To further explore the effect of pressure on *K* and *G*, the pressure derivatives *K′* and *G′* were calculated based on linear fitting

results of *K* and *G* with pressure, respectively. The *K′* values of CaC$_2$O$_5$s fall within the range of 3.8 ± 0.3 indicating that the *K* values of the 6 polymorphs exhibit a similar trend with pressure. The *G′* of CaC$_2$O$_5$-*C2* has a maximum value of 1.7, and the other 5 polymorphs of CaC$_2$O$_5$ have similar values of 1.2 ± 0.1. *K* and *G* of CaC$_2$O$_5$-*I$\bar{4}$2d* and CaC$_2$O$_5$-*C2-l* show very similar values under high pressure, indicating that they have the same elastic properties.

Generally, understanding the composition, physical state, and structure of the Earth's interior mainly relies on observing the seismic wave velocity. Quantified seismic velocity data link seismic observations and geological characteristics, such as modal mineral composition and velocity structure. Laboratory measurements and computer simulations of the seismic wave velocity of minerals under appropriate pressure and temperature conditions have long been used to interpret the velocity structure of the Earth, combined with the results of

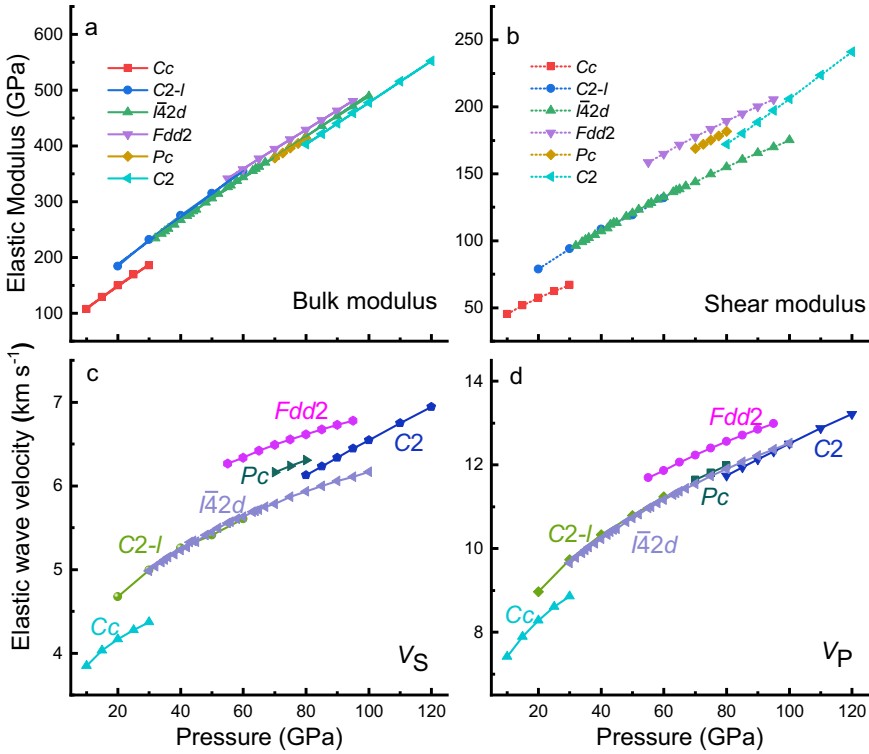

**Fig. 3 | Elastic modulus and elastic wave velocity of CaC$_2$O$_5$s. a** Bulk modulus (solid line) and **b** shear modulus (point line) of CaC$_2$O$_5$ (the space group symbol is shown in the figure) within their respective structural phase transformation pressures[5]. **c** Shear-wave velocity ($V_S$) and **d** compressional-wave velocity ($V_P$) of CaC$_2$O$_5$s within their respective structural phase transformation pressures[5].

large-scale geophysical imaging techniques[27]. As a potentially important component of the deep Earth, the seismic wave velocity of CaC$_2$O$_5$ with different symmetries under high pressure is highly important for understanding the structure and composition of the mantle. Therefore, we can calculate the shear-wave ($V_S$) and compressional-wave ($V_P$) velocities of CaC$_2$O$_5$s (see Methods) in Fig. 3c, d.

CaC$_2$O$_5$-$Cc$ has the slowest wave velocity, and CaC$_2$O$_5$-$Fdd2$ has the largest wave velocity among the 6 polymorphs. As the pressure increases, the wave velocities of CaC$_2$O$_5$s increase. For 6 CaC$_2$O$_5$ polymorphs, the $V_S$ values are sorted as $Fdd2 > Pc > C2 > I\bar{4}2d \approx C2$-$l > Cc$, and the $V_P$ values are sorted as $Fdd2 > Pc > I\bar{4}2d \approx C2$-$l > C2 > Cc$. The wave velocities of CaC$_2$O$_5$-$I\bar{4}2d$ and CaC$_2$O$_5$-$C2$-$l$ are very close.

### Relationship between CaC$_2$O$_5$-$C2$-$l$ and CaC$_2$O$_5$-$I\bar{4}2d$

As discussed above, some of the characteristics of CaC$_2$O$_5$-$C2$-$l$ and CaC$_2$O$_5$-$I\bar{4}2d$ show good consistency. For example, under the same pressure, the lattice parameters $a$ and $b$, and the density of these two polymorphs are almost the same (the difference is less than 0.02), and the lattice parameter $c$ maintains a constant ratio of 0.8 (Supplementary Fig. 2). The differences in the lattice parameter $c$ of the two polymorphs came from the differences in the bond angle $\beta$ in their cell structure, namely, 125° in CaC$_2$O$_5$-$C2$-$l$ and 90° in CaC$_2$O$_5$-$I\bar{4}2d$. The $V_P$ and $V_S$ of CaC$_2$O$_5$-$C2$-$l$ and CaC$_2$O$_5$-$I\bar{4}2d$ also exhibit good consistency under the same pressure (the differences in $V_P$ and $V_S$ are less than 0.08 km s$^{-1}$ and 0.03 km s$^{-1}$, respectively) (Fig. 4).

To further verify the relationship between the CaC$_2$O$_5$-$C2$-$l$ and CaC$_2$O$_5$-$I\bar{4}2d$, we calculated the energy band structure and the density of states under high pressure from CaC$_2$O$_5$s.

The band gap of CaC$_2$O$_5$s varies between 5.108 eV and 7.701 eV under the explored pressure in this work, indicating their insulating properties[28] (Fig. 4a). The band gaps of CaC$_2$O$_5$-$C2$-$l$ and CaC$_2$O$_5$-$I\bar{4}2d$ are almost the same at the same pressure (the band gap difference is

less than 0.02 eV). As the pressure increases, the bandgaps of CaC$_2$O$_5$-$C2$-$l$ and CaC$_2$O$_5$-$I\bar{4}2d$ gradually increase; however, the bandgaps of the other 4 polymorphs, CaC$_2$O$_5$-$Cc$, CaC$_2$O$_5$-$Pc$, CaC$_2$O$_5$-$Fdd2$, and CaC$_2$O$_5$-$C2$, decrease with increasing pressure. Under 10 to 50 GPa, the relative conductivity of CaC$_2$O$_5$s follows the order CaC$_2$O$_5$-$Cc$ > CaC$_2$O$_5$-$C2$-$l$ ≈ CaC$_2$O$_5$-$I\bar{4}2d$; when the pressure is greater than 50 GPa, the relative conductivity of CaC$_2$O$_5$s is as follows the order CaC$_2$O$_5$-$C2$ > CaC$_2$O$_5$-$Fdd2$ > CaC$_2$O$_5$-$Pc$ > CaC$_2$O$_5$-$I\bar{4}2d$.

The electronic density of states (DOS) and the band structure of CaC$_2$O$_5$-$I\bar{4}2d$ and CaC$_2$O$_5$-$C2$-$l$ are nearly identical under the same pressure range (Fig. 4b, c). The difference in the orbitals contributions is less than 0.3%. The characteristics of the DOS of CaC$_2$O$_5$-$C2$-$l$ and CaC$_2$O$_5$-$I\bar{4}2d$ are primarily determined by the electrons distributed in the p orbitals. At 40 GPa, the p orbitals contribute approximately 62.94% and 63.12%, respectively, to the total DOS. Moreover, the remaining contributions are attributed to the s orbitals (23.67% and 23.48%) and the d orbitals (13.39% and 13.40%).

The cell parameters, electronic properties, elasticity, and wave velocity of CaC$_2$O$_5$-$C2$-$l$ and CaC$_2$O$_5$-$I\bar{4}2d$ under high pressure are almost identical, and the trends with pressure are also almost identical. Hence, these two phases are the same thing.

### Effects of CaC$_2$O$_5$s on the structure of the mantle

To explore the effects of CaC$_2$O$_5$ polymorphs on the structure of the mantle, we compared our calculations with the wave velocities of sseveral common mantle minerals (Fig. 5), including wadsleyite[29], ringwoodite[30], cubic Ca-Pv[31], MgSiO$_3$ perovskite[18] and CaSiO$_3$ perovskite[17] under high pressure. Moreover, the seismic wave velocity of the PREM[22] is also presented here.

The wave velocities of CaC$_2$O$_5$s under the explored high pressure in this work are lower than those of the PREM model. The wave velocities of CaC$_2$O$_5$s are also lower than those of common minerals in the

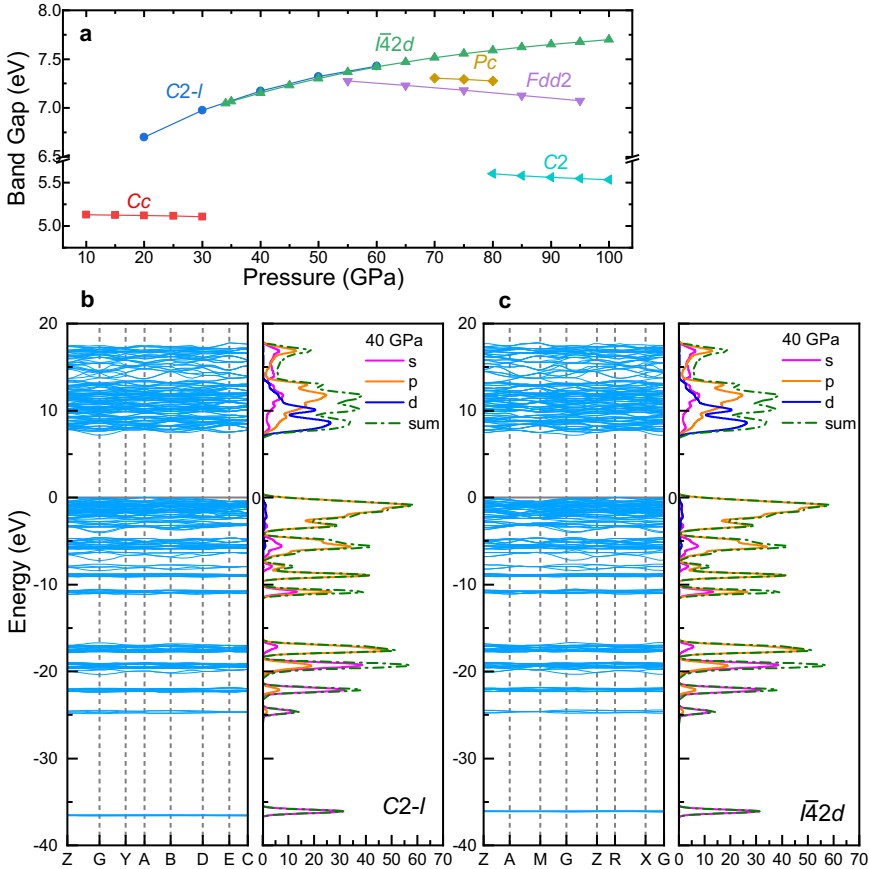

**Fig. 4 | Electronic properties of CaC₂O₅s. a** The band gap of CaC₂O₅s (the space group symbol is shown in the figure) within their respective structural transformation pressures. **b**, **c** The band structure and density of states of CaC₂O₅-*C2-l* and CaC₂O₅-*Ī42d* at 40 GPa. Blue lines are the band structures, the magenta, orange, and dark blue line represent the contributions of s, p, and d orbitals to the density of states, respectively, and the dark green dot lines represent the total density of states.

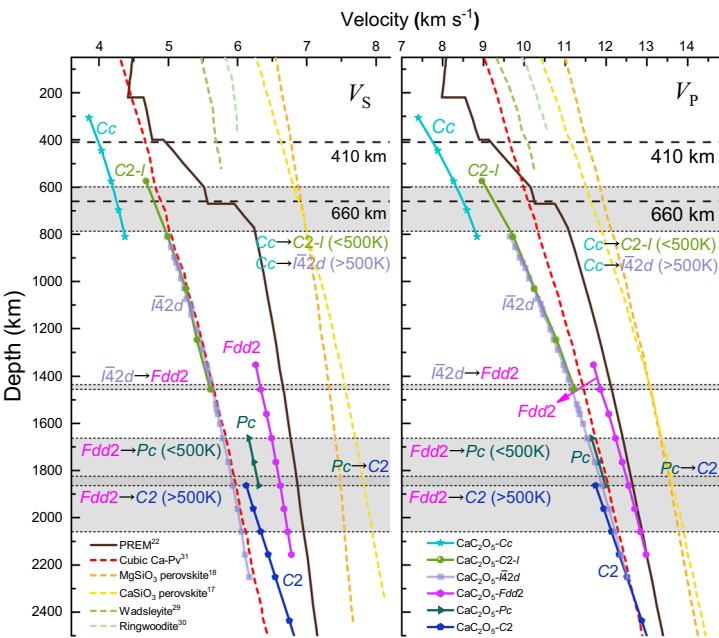

**Fig. 5 | Shear-wave velocity ($V_S$) and compressional-wave velocity ($V_P$) of CaC₂O₅s, the PREM model, and some mantle minerals.** Comparison of the seismic wave velocity of wadsleyite, ringwoodite, cubic Ca-Pv[31], MgSiO₃ perovskite[18] and CaSiO₃ perovskite[17] and CaC₂O₅s (the space group symbol is shown in the figure) with that of Earth's mantle according to the PREM[22], the black thin dotted line, the gray area, and the black arrows represent the boundaries at which the phase transition begins, the region where the phase transition occurs, and which two types of CaC₂O₅ undergo the phase transition[5], respectively.

mantle, including wadsleyite, ringwoodite, $MgSiO_3$ perovskite, and $CaSiO_3$ perovskite; therefore $CaC_2O_5$s are a set of low-velocity minerals and may be useful for understanding the origin of the low-velocity layer in the mantle.

From depths of 300 km to 1662 km, the wave velocity increases with pressure. Two significant velocity surges are observed during the phase transformation among the $CaC_2O_5$ polymorphs: one is near 660 km depth when the $CaC_2O_5$-$Cc$ structure transforms into $CaC_2O_5$-$I\bar{4}2d$ ($C2$-$l$), and the other is near 1435 km depth when the $CaC_2O_5$-$I\bar{4}2d$ structure transforms into $CaC_2O_5$-$Fdd2$. Afterwards, from depths of 1662 km to 2500 km, there is a sudden drop in wave velocity when $CaC_2O_5$-$Fdd2$ transforms into $CaC_2O_5$-$C2$ and $CaC_2O_5$-$Pc$ near a depth of 1662 km.

$CaC_2O_5$-$Cc$ transforms into $CaC_2O_5$-$I\bar{4}2d$ ($CaC_2O_5$-$C2$-$l$) at 21 GPa (0 K) to 29 GPa (2500 K)[5], leading to an increase in $V_S$ of 13.84% and an increase in $V_P$ of 9.18% (Fig. 5). This phase transformation depth matches the depth of the 660 km discontinuity zone of the mantle[32,33]. The general view of the seismic wave velocity discontinuity at 660 km in the mantle transition zone is caused by the postspinel phase transformation of ringwoodite (ringwoodite (Mg, Fe)2SiO₄ → perovskite (Mg, Fe)SiO₃ + ferropericlase (Mg, Fe)O)[34]. The density of the PREM model increases from 3.99 g cm⁻³ to 4.38 g cm⁻³, $V_S$ increases from 5.57 km s⁻¹ to 5.95 km s⁻¹, and $V_P$ increases from 10.26 km s⁻¹ to 10.75 km s⁻¹ at 660 km, and the increase in density, $V_S$ and $V_P$ are 0.37 g cm⁻³, 0.38 km s⁻¹, and 0.49 km s⁻¹, respectively. When $CaC_2O_5$-$Cc$ transforms into $CaC_2O_5$-$I\bar{4}2d$ ($CaC_2O_5$-$C2$-$l$) at a depth of 660 km, its density increases from 3.38 g cm⁻³ to 3.67 g cm⁻³ by 0.29 g cm⁻³, $V_S$ increases from 4.25 km s⁻¹ to 4.81 km s⁻¹ by 0.55 km s⁻¹, and $V_P$ increases from 9.28 km s⁻¹ to 8.50 km s⁻¹ by 0.78 km s⁻¹. The $CaC_2O_5$ and PREM have very similar densities, and the shear-wave velocity increases at 660 km. Therefore, we propose that the transformation of $CaC_2O_5$-$Cc$ to $CaC_2O_5$-$I\bar{4}2d$ ($CaC_2O_5$-$C2$-$l$) may also be one of the origins of wave velocity discontinuity at a depth of 660 km in the mantle transition zone.

Studying the wave velocity structure of the large low-shear-velocity provinces (LLSVPs) in the lower mantle is highly important for understanding the dynamic evolution process of Earth's interior. LLSVPs play a crucial role in regulating heat flow from the core, facilitating the exchange of matter and energy between the Earth's deep layers and influencing continental evolution, surface resources, and the environment. LLSVPs are regions with shear wave velocities ($V_S$) several percent lower than those of the surrounding mantle[35–38]. There are two large low shear wave velocity bodies that extend thousands of kilometres horizontally and hundreds of kilometres vertically on the Core-Mantle boundary below Africa and the Pacific Ocean[38–40]. There are many controversies about the origin and evolution of these areas[41]. Thomson et al. proposed a cubic Ca-Pv perovskite that can perfectly match the characteristics of low-shear-velocity minerals in LLSVPs[31] (Fig. 5) and is considered an interpretation of the origin of LLSVPs. Here, our calculated $V_S$ of $CaC_2O_5$-$I\bar{4}2d$ is almost the same as that of cubic Ca-Pv perovskite, and when $CaC_2O_5$-$Fdd2$ transforms into $CaC_2O_5$-$C2$ as the pressure increases, its $V_S$ decreases by 7.4% (Fig. 5). It's worth noting that $CaC_2O_5$-$I\bar{4}2d$ was synthesized via high-temperature and high-pressure experiments at 34 GPa and 45 GPa by the reaction of $CaCO_3 + CO_2$[4]. Moreover, Yao et al. proposed[3] that $CaC_2O_5$ might react with perovskite (Mg, Ca)$SiO_3$ and ferropericlase MgO in the deep mantle to generate $MgCO_3$, $CaCO_3$, $CO_2$, and perovskite (Mg, Ca)$SiO_3$, etc. LLSVPs not only exhibit low-velocity velocities that are several percent lower than the those of the surrounding mantle but also exhibit higher density than the surrounding components[42–44]. When $CaC_2O_5$-$Fdd2$ transforms into $CaC_2O_5$-$C2$, its density increases by 5.8% (Fig. 1). This trend allows $CaC_2O_5$ transformation to more perfectly match the characteristics of LLSVPs. Therefore, we determine that $CaC_2O_5$-$I\bar{4}2d$ and its high-pressure polymorphs are very likely to be in the interior of the LLSVPs region and may serve

as a supplement to the Ca-Pv component in LLSVPs or may be one of the main components in LLSVPs.

Based on our calculated density, wave velocity of $CaC_2O_5$s, probable phase transformation, and reactions of $CaC_2O_5$ in the mantle, a model of the deep carbon cycle is proposed (Fig. 6). Calcium carbonate ($CaCO_3$) and $CO_2$ can reach the deep Earth through subduction plates. The discovery of calcium carbonate inclusions in superdeep diamonds[45] and fluid $CO_2$ wrapped in diamonds[46] also confirmed the occurrence of calcium carbonate and $CO_2$ in the deep mantle. Therefore, $CaC_2O_5$ and its high-pressure polymorphs could be produced by the reaction $CaCO_3 + CO_2 \rightarrow CaC_2O_5$ at different depths[4] in the mantle. $CaC_2O_5$-$Cc$ in the shallow mantle transforms into $CaC_2O_5$-$I\bar{4}2d$ at 660 km depth, causing an anomalous increase in wave velocity in the mantle transition zone. Meanwhile, $CaC_2O_5$-$Fdd2$ in the deep mantle may partially transforms into $CaC_2O_5$-$C2$ with increasing pressure, leading to an increase in mantle density but a decrease in shear-wave velocity, which is one of the possible reasons for explaining the origin of LLSVPs[42–44]. During these reactions, $CaC_2O_5$ reacts with minerals such as (Mg, Ca)$SiO_3$ perovskite and ferropericlase (MgO) in the lower mantle and produce magnesium carbonate ($MgCO_3$), calcium carbonate ($CaCO_3$), calcium silicate($CaSiO_3$), and $CO_2$[3,5]. The more stable $MgCO_3$ and $CaSiO_3$ remain in the lower mantle as the main carbonate minerals[3] and the main minerals in LLSVPs[31], respectively. $CaCO_3$ generated by the reaction may repeat the process of reacting with $CO_2$ to generate $CaC_2O_5$ or decompose into $CO_2$[3,4]. $CO_2$ and carbon-bearing minerals can also be transported to the Earth's shallow through

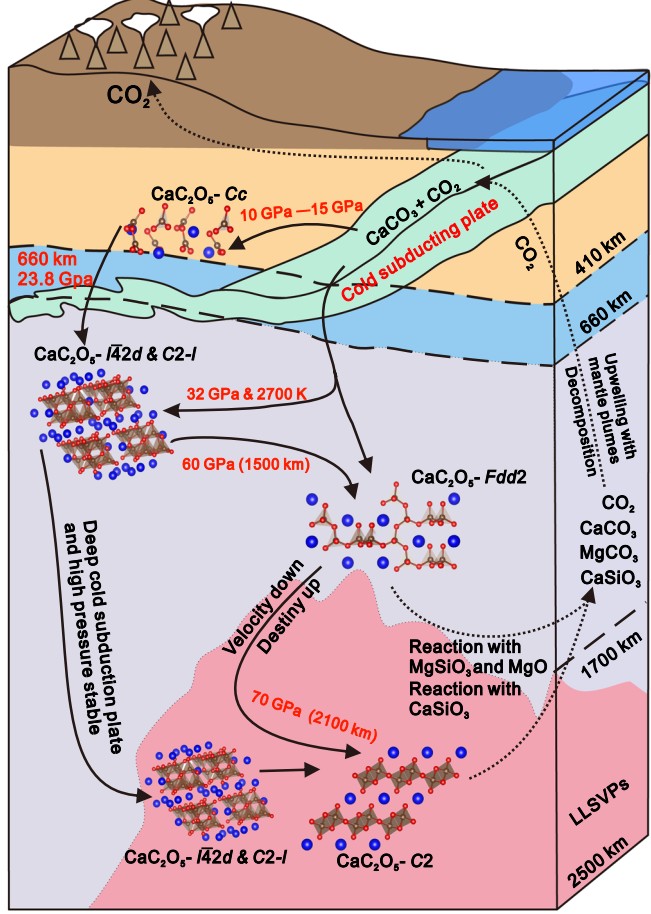

**Fig. 6 | Schematic diagram of $CaC_2O_5$s in the mantle.** Crystal structure of $CaC_2O_5$s and its possible reactions and reaction environments in the mantle are described. Solid arrows indicate the reaction or phase transition that generates $CaC_2O_5$ (the red letter part indicates the phase transition conditions[45]), while dashed arrows indicate the reaction that generates other substances[3].

internal dynamic processes. For example, mantle plumes, which are high-temperature environments, facilitate the rise of gaseous carbon dioxide back to the upper mantle and the bottom of the lithosphere. Therefore, $CaC_2O_5$ may play a significant role in Earth's deep carbon cycle as follows: $CaC_2O_5$ would transports carbon to the deep Earth while it also undergoes successive structural phase transformations with increasing pressure, after which $CO_2$ and carbon-bearing minerals are produced via reactions of $CaC_2O_5$ with minerals that transport carbon to shallow parts of the Earth.

The structural, electronic, and elastic properties of $CaC_2O_5$s, including those of $CaC_2O_5$-$Cc$, $CaC_2O_5$-$Fdd2$, $CaC_2O_5$-$C2$-$l$, $CaC_2O_5$-$C2$, $CaC_2O_5$-$I\bar{4}2d$, and $CaC_2O_5$-$Pc$, were studied under high pressure by first-principles simulation. The calculations show that $CaC_2O_5$ is a group of low-density mantle minerals with a low seismic wave velocity. The $CaC_2O_5$-$I\bar{4}2d$ and $CaC_2O_5$-$C2$-$l$ exhibit good agreement in their crystal cell structures, electronic state densities, band structures, and seismic wave velocities, indicating that these two polymorphs can be considered the same.

Changes in seismic wave velocity caused by phase transformation of the $CaC_2O_5$-Cc to $CaC_2O_5$-$I\bar{4}2d$ ($CaC_2O_5$-$C2$-$l$) agree with the wave velocity discontinuity at a depth of 660 km in the mantle transition zone and maybe one of the origins of this discontinuity. Moreover, when $CaC_2O_5$-$Fdd2$ transforms into $CaC_2O_5$-$C2$, its $V_S$ decreases by 7.4%, and its density increases by 5.8%, perfectly matching the characteristics of LLSVPs. As the pressure increased, $CaC_2O_5$ eventually transformed into $CaC_2O_5$-$C2$. Generally, $CaC_2O_5$ and its high-pressure polymorphs may be among the main components of LLSVPs.

Therefore, $CaC_2O_5$ exhibits important effects on the structure and composition of the mantle as well as the Earth's deep carbon cycle. Our work provides different insights for understanding the origin of the wave velocity discontinuity at the depth of 660 km in the mantle transition zone and at the LLSVPs. It also sheds light on the deep carbon cycle model of the Earth.

## Methods

### The density functional theory (DFT) calculations

First principles calculations are performed using density functional theory[47,48] with the plane-wave pseudopotential. The calculations are implemented in the CASTEP code[49]. The generalized gradient approximation (GGA) with PBE parameterization[50] is employed to describe exchange-correlation interactions. OTFG ultrasoft pseudopotentials[51,52] are utilized to model electron-ion interactions with a plane-wave energy cutoff set in Table 1. A Monkhorst Pack grid of k-points was employed to sample the Brillouin zone. Table 1 presents the plane-wave energy cutoff and k-points settings for the various $CaC_2O_5$ phase states. A convergence criterion of $5 \times 10^{-7}$ a.u. for total energy is employed for self-consistent-field calculations.

$CaC_2O_5$s under different pressures are calculated by simultaneously optimizing atomic positions and lattice constants, respectively, using Hellmann-Feynman forces and stresses acting on nuclei and lattice parameters[53]. The phonon mode is determined by finite displacement calculations[54] to ensure molecular stability. Bonding characteristics are determined using Mulliken's population analysis[55,56]. Stress-strain relations calculate elastic constants[23], where all applied strains have magnitudes of 0.003, and the linear relation was ensured to be sufficient for this strain range. Density of states and partial density of states also caculated by CASTEP code[49]. Crystal structures were visualized with VESTA[57].

### Benchmark calculation

The lattice parameters and cell volume of $CaC_2O_5$s were calculated and compared with reported values to evaluate the accuracy of the density functional theory approach employed here. Differences between our calculated lattice parameters and volume and previous simulation are less than 1.24%, and the difference with experimental results is 2.98% (Table 2), demonstrating the accuracy of our calculation scheme.

**Table 1 | The plane-wave energy cutoff and k-points settings of $CaC_2O_5$**

| $CaC_2O_5$ phase | Energy cutoff | K-points | $CaC_2O_5$ phase | Energy cutoff | K-points |
|---|---|---|---|---|---|
| $CaC_2O_5$-$C2$ | 1000 eV | 6 × 7 × 5 | $CaC_2O_5$-$Fdd2$ | 1020 eV | 3 × 6 × 6 |
| $CaC_2O_5$-$C2$-$l$ | 1000 eV | 3 × 4 × 5 | $CaC_2O_5$-$I\bar{4}2d$ | 1020 eV | 4 × 4 × 3 |
| $CaC_2O_5$-$Cc$ | 1010 eV | 5 × 3 × 8 | $CaC_2O_5$-$Pc$ | 1000 eV | 8 × 9 × 7 |

**Table 2 | Structure and volume of $CaC_2O_5$s**

| Structural phase | Pressure (GPa) | Lattice Parameters | | | | Volume | | Reference |
|---|---|---|---|---|---|---|---|---|
| | | a/Å | b/Å | c/Å | Angle β/deg | Å³/f.u. | Δ | |
| $CaC_2O_5$-$C2$ | 100 | 6.891 | 3.231 | 9.073 | 150.450 | 99.627 | | This study |
| | | 6.902 | 3.235 | 9.090 | 150.5 | 99.943 | 0.31% | Yao et al.[3] |
| $CaC_2O_5$-$C2$-$l$ | 50 | 11.999 | 6.925 | 6.925 | 125.242 | 469.963 | | This study |
| | | 12.019 | 6.936 | 6.936 | 125.250 | 472.191 | 0.47% | Sagatova et al.[5] |
| $CaC_2O_5$-$Cc$ | 15 | 7.425 | 10.415 | 4.541 | 121.197 | 300.448 | | This study |
| | | 7.435 | 10.421 | 4.589 | 121.190 | 304.162 | 1.24% | Sagatova et al.[5] |
| $CaC_2O_5$-$Fdd2$ | 60 | 13.972 | 5.607 | 5.662 | 90 | 443.591 | | This study |
| | | 13.990 | 5.615 | 5.671 | 90 | 445.479 | 0.43% | Yao et al.[3] |
| $CaC_2O_5$-$I\bar{4}2d$ | 34 | 7.063 | 7.063 | 10.002 | 90 | 498.977 | | This study |
| | X-ray | 6.994 | 6.994 | 9.897 | 90 | 484.1 | −2.98% | König et al.[4] |
| | DFT | 7.054 | 7.054 | 9.989 | 90 | 497 | −0.40% | König et al.[4] |
| $CaC_2O_5$-$Pc$ | 80 | 4.537 | 3.829 | 6.061 | 94.966 | 104.883 | | This study |
| | | 4.545 | 3.834 | 6.069 | 95 | 105.353 | 0.45% | Yao et al.[3] |

Compared our calculation results with those of previous researchers under the same pressure, including lattice constant, lattice β Angle and cell volume, Δ is the deviation between the our calculated volume and previous results.

Interestingly, lattice parameters and cell volume of $CaC_2O_5$-$I\bar{4}2d$ from Sagatova[5] differ significantly from our simulated results ($\Delta = 10.68\%$). The values also differ from the experimental result[4]. Sagatova[5] calculated the lattice parameter at 50 GPa[5], which is larger than the experimental value obtained at 34 GPa[4].

All crystal structure files of $CaC_2O_5$s used in this study can be obtained from the supplementary files (Supplementary Table 2 and Supplementary Fig. 3).

Formula for calculating the velocity of the $V_S$ and $V_P$

$$V_S = \sqrt{G/\rho} \tag{1}$$

$$V_P = \sqrt{(K + 4G/3)/\rho} \tag{2}$$

Where $K$ (GPa) is the bulk modulus, $G$ (GPa) is the shear modulus, and $\rho$ (g cm$^{-3}$) is the density.

## Data availability

The authors declare that the main data supporting the findings of this study are contained within the paper and its associated Supplementary Information. The raw data obtained from this paper calculation are available at figshare[58] (https://doi.org/10.6084/m9.figshare.24948159). All other relevant data are available from the corresponding authors upon request.

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

## Acknowledgements

This work was supported by the National Natural Science Foundation of China (42174115, 42330311 and 42073056), the Special Fund of the Institute of Earthquake Forecasting, China Earthquake Administration (CEAIEF20230301), and the State key laboratory of earthquake dynamics (Project No. LED2021B02).

## Author contributions

H.W., L.L. and S.M. contributed to conception and design of this study. H.W. built the model and calculated the data and plot them. H.W. and L.L. wrote the paper. H.W. and Z.G. constructed the stratigraphic model diagram describing the content of the article. H.W., L.Y. and G.N. contributed to the data collection. All authors contributed to the discussion and revision of the paper.

## Competing interests

The authors declare no competing interests.
