## [Peer Review File · Nature Communications]

REVIEWER COMMENTS

Reviewer #1 was withdrawn from the process when they became unable to supply a report

Reviewer #2 (Remarks to the Author):

The study of Wang et al. is devoted to the calculation within the density functional theory of structural, electronic and elastic properties of known CaC₂O₅ polymorphs. The authors proposed an interesting scenario of the possible influence of CaC₂O₅ modifications on the explanation of the seismic velocity anomalies in the Earth's mantle. The technical part of the work is not in doubt and is done correctly. The scientific part is well founded. I recommend it to publish after considering the following points.

The manuscript is written a little chaotically with many grammatical and stylistic errors (see lines 15, 91, 230), which leaves a rather ambiguous impression. The quality of English should be improved (needs polishing).

The article often contains the phrase "6 CaC₂O₅ polymorphs", it seems to me absolutely superfluous to write like that (enough to mention it in the introduction).

In the text, the authors state that electrical properties have been studied. This is an incorrect statement. Electrical properties are usually understood as conductivity, resistance etc. Probably, the authors meant electronic.

It is necessary to carefully review the figures, especially figure 9, which contains some typos and errors in the text.

Formula (1) must be moved to the Methods section.

Reviewer #3 (Remarks to the Author):

Carbonate represents a crucial constituent of the Earth and exerts a significant influence on the structural characteristics of the Earth's interior. Recently discovered CaC₂O₅-minerals have been identified as potential components of the mantle; however, a comprehensive understanding of their properties remains elusive, hindering a deeper comprehension of their impact on the Earth's interior composition and structure. This manuscript elucidates the structural, elastic, and electrical properties of six distinct CaC₂O₅ phases under high pressure through first principles simulations. The findings contribute essential insights into the phase diagram of CaC₂O₅ under conditions relevant to the Earth's mantle and its associated properties. Notably, the seismic wave velocity alterations resulting from the phase transition of CaC₂O₅-Cc to CaC₂O₅-I2d (CaC₂O₅-C2-I) align well with the wave velocity discontinuity observed at a depth of 660 km in the mantle transition zone. Furthermore, under higher pressure, CaC₂O₅-Fdd2 can transition to CaC₂O₅-C2 at depths exceeding 2100 km (70 GPa), resulting in a 7.4% reduction in V_s and a 5.8% increase in density. This observation may elucidate the characteristics of the observed Large Low Shear Velocity Provinces (LLSVPs).

The manuscript introduces novel perspectives for understanding the origin of the wave velocity discontinuity at 660 km and the LLSVPs based on the properties of CaC₂O₅-minerals. This work is anticipated to garner widespread attention in the field of Earth science. I recommend publishing this work in Nature Communications after some minor modifications.

Several questions and suggestions are provided:

1. According to the PREM, the pressure at the depth of 2900km shall be about 135GPa, however, in the figure 8, the pressure at the 2900 is about 100GPa. Discrepancies in the pressure-depth relationship are noted leads to inconsistencies in Figures 3 and 8. While this discrepancy does not impact the conclusions, corrections are advised.
2. The conversion of CaC₂O₅-I42d to CaC₂O₅-Fdd2 results in an increase in seismic wave velocity, yet the author does not discuss the implications of this change on mantle structure or composition. This aspect requires further exploration.
3. The benchmark calculation should be included in the method section.
4. Several corrections are recommended, such as modifying table and figure references, addressing typos, and ensuring proper formatting of journal names in the references.

Line 68: Table 2 should be Table 1;

Figure 9: Ca₂CO₅ should be CaC₂O₅;

Line 74.75: reference 4.5 already cited in line 73 and not needed again

Line 78: "two pressure" should be "two pressures"

Line 116: "polymorphs is is constant" should be "polymorphs is constant"

Line 123: "δ-AlOOH₂O and" should be "δ-AlOOH₂O, and"

Line 240: PREM model should be the PREM model

Line 259: "at the depth" should be "at a depth"

Line 283: delete "and"

Line 327: density should be density

Figure 9: Ca₂CO₅ should be CaC₂O₅

Incorrect way of reference listed of References 2;

REVIEWER COMMENTS

Reviewer #1 was withdrawn from the process when they became unable to supply a report

Reviewer #2 (Remarks to the Author):

The study of Wang et al. is devoted to the calculation within the density functional theory of structural, electronic and elastic properties of known CaC₂O₅ polymorphs. The authors proposed an interesting scenario of the possible influence of CaC₂O₅ modifications on the explanation of the seismic velocity anomalies in the Earth's mantle. The technical part of the work is not in doubt and is done correctly. The scientific part is well founded. I recommend it to publish after considering the following points.

The manuscript is written a little chaotically with many grammatical and stylistic errors (see lines 15, 91, 230), which leaves a rather ambiguous impression. The quality of English should be improved (needs polishing).

The article often contains the phrase “6 CaC₂O₅ polymorphs” , it seems to me absolutely superfluous to write like that (enough to mention it in the introduction).

In the text, the authors state that electrical properties have been studied. This is an incorrect statement. Electrical properties are usually understood as conductivity, resistance etc. Probably, the authors meant electronic.

It is necessary to carefully review the figures, especially figure 9, which contains some typos and errors in the text.

Formula (1) must be moved to the Methods section.

Reviewer #3 (Remarks to the Author):

Carbonate represents a crucial constituent of the Earth and exerts a significant influence on the structural characteristics of the Earth's interior. Recently discovered CaC₂O₅-minerals have been identified as potential components of the mantle; however, a comprehensive understanding of their properties remains elusive, hindering a deeper comprehension of their impact on the Earth's interior composition and structure. This manuscript elucidates the structural, elastic, and electrical properties of six distinct CaC₂O₅ phases under high pressure through first principles simulations. The findings contribute essential insights into the phase diagram of CaC₂O₅ under conditions relevant to the Earth's mantle and its associated properties. Notably, the seismic wave velocity alterations resulting from the phase transition of

CaC₂O₅-Cc to CaC₂O₅-I2d (CaC₂O₅-C2-1) align well with the wave velocity discontinuity observed at a depth of 660 km in the mantle transition zone. Furthermore, under higher pressure, CaC₂O₅-Fdd2 can transition to CaC₂O₅-C2 at depths exceeding 2100 km (70 GPa), resulting in a 7.4% reduction in V_s and a 5.8% increase in density. This observation may elucidate the characteristics of the observed Large Low Shear Velocity Provinces (LLSVPs).

The manuscript introduces novel perspectives for understanding the origin of the wave velocity discontinuity at 660 km and the LLSVPs based on the properties of CaC₂O₅-minerals. This work is anticipated to garner widespread attention in the field of Earth science. I recommend publishing this work in Nature Communications after some minor modifications.

Several questions and suggestions are provided:

1. According to the PREM, the pressure at the depth of 2900km shall be about 135GPa, however, in the figure 8, the pressure at the 2900 is about 100GPa. Discrepancies in the pressure-depth relationship are noted leads to inconsistencies in Figures 3 and 8. While this discrepancy does not impact the conclusions, corrections are advised.
2. The conversion of CaC₂O₅-I42d to CaC₂O₅-Fdd2 results in an increase in seismic wave velocity, yet the author does not discuss the implications of this change on mantle structure or composition. This aspect requires further exploration.
3. The benchmark calculation should be included in the method section.
4. Several corrections are recommended, such as modifying table and figure references, addressing typos, and ensuring proper formatting of journal names in the references.

Line 68: Table 2 should be Table 1;

Line 74.75: reference 4.5 already cited in line 73 and not needed again

Line 78: “two pressure” should be “two pressures”

Line 116: “polymorphs is is constant” should be “polymorphs is constant”

Line 123: “6-AlOOH20 and” should be “6-AlOOH20, and”

Line 240: PREM model should be the PREM model

Line 259: “at the depth” should be “at a depth”

Line 283: delete “and”;

Figure 9: Ca₂CO₅ should be CaC₂O₅;

Line 327: densitiy should be density

Incorrect way of reference listed of References 2;

On behalf of all the contributing authors, we appreciate editor and reviewers very much for their positive and constructive comments and suggestions on our manuscript. These comments are all valuable and helpful for improving our article. According to the editor and reviewers' comments, we have made extensive modifications to our manuscript. In this revised version, changes to our manuscript were all highlighted within the document by using red-colored text and turquoise background. The reviewer comments are laid out below in *italicized font* and specific concerns have been numbered. Our response is given in normal font.

Response to the Reviewer #2

The study of Wang et al. is devoted to the calculation within the density functional theory of structural, electronic and elastic properties of known CaC₂O₅ polymorphs. The authors proposed an interesting scenario of the possible influence of CaC₂O₅ modifications on the explanation of the seismic velocity anomalies in the Earth's mantle. The technical part of the work is not in doubt and is done correctly. The scientific part is well founded. I recommend it to publish after considering the following points.

A: The authors agree with the assessment of the paper, and it accurately reflects the ideas espoused in this study.

The manuscript is written a little chaotically with many grammatical and stylistic errors (see lines 15, 91, 230), which leaves a rather ambiguous impression. The quality of English should be improved (needs polishing).

A: We are very sorry for the mistakes in this manuscript and inconvenience they caused in your reading. We have corrected the grammatical errors in the resubmitted manuscript version (line 16-19, line 82-83 and line 225). We have thoroughly polished and corrected the language and formatting of our manuscript using **Research Square AJE Services** to ensure the correctness of the language and formatting of our article, so we hope it can meet the journal's standard and all change highlight in red.

The article often contains the phrase "6 CaC₂O₅ polymorphs", it seems to me absolutely superfluous to write like that (enough to mention it in the introduction).

A: Regarding the usage of "6 CaC₂O₅ polymorphs," we appreciate your correction and have rephrased this expression in "CaC₂O₅s" throughout the entire article (mention it in the introduction at line 59 and highlight in red).

In the text, the authors state that electrical properties have been studied. This is an incorrect statement. Electrical properties are usually understood as conductivity, resistance etc. Probably, the authors meant electronic.

A: As suggested by the reviewer, we have corrected the “electrical” into “electronic” throughout the entire article and highlighted all changes in red.

It is necessary to carefully review the figures, especially figure 9, which contains some typos and errors in the text.

A: We have corrected the error “Ca₂CO₅” to “CaC₂O₅” in Fig. 9 and further checked the icons and references throughout the entire article. All changes have been highlighted in red.

Formula (1) must be moved to the Methods section.

A: Based on your and another reviewer’ s comments, we have moved Formula (1) and The benchmark calculation to the Methods section of the article (line 365, line 347). And we thank the reviewer for raising these pertinent issues.

Response to the Reviewer #3

Carbonate represents a crucial constituent of the Earth and exerts a significant influence on the structural characteristics of the Earth's interior. Recently discovered CaC₂O₅-minerals have been identified as potential components of the mantle; however, a comprehensive understanding of their properties remains elusive, hindering a deeper comprehension of their impact on the Earth's interior composition and structure. This manuscript elucidates the structural, elastic, and electrical properties of six distinct CaC₂O₅ phases under high pressure through first principles simulations. The findings contribute essential insights into the phase diagram of CaC₂O₅ under conditions relevant to the Earth's mantle and its associated properties. Notably, the seismic wave velocity alterations resulting from the phase transition of CaC₂O₅-Cc to CaC₂O₅-I42d (CaC₂O₅-C2-l) align well with the wave velocity discontinuity observed at a depth of 660 km in the mantle transition zone. Furthermore, under higher pressure, CaC₂O₅-Fdd2 can transition to CaC₂O₅-C2 at depths exceeding 2100 km (70 GPa), resulting in a 7.4% reduction in V_s and a 5.8% increase in density. This observation may elucidate the characteristics of the observed Large Low Shear Velocity Provinces (LLSVPs).

The manuscript introduces novel perspectives for understanding the origin of the wave velocity discontinuity at 660 km and the LLSVPs based on the properties of CaC₂O₅-minerals. This work is anticipated to garner widespread attention in the field of Earth science. I recommend publishing this work in Nature Communications after some minor modifications.

A: We thank the reviewer for the positive assessment of our reported work.

Several questions and suggestions are provided:

1. According to the PREM, the pressure at the depth of 2900 km shall be about 135 GPa, however, in the figure 8, the pressure at the 2900 is about 100 GPa. Discrepancies in the pressure-depth relationship are noted leads to inconsistencies in Figures 3 and 8. While this discrepancy does not impact the conclusions, corrections are advised.

A: We are sorry for our carelessness. Based on your comments, we have corrected the pressure and depth conversion in the PREM model, rectified the errors in Fig. 3 and Fig. 8, and made corrections to the data using a new conversion ratio. The PREM data are given in the Table 1. The pressure data between two adjacent depths is calculated using the linear interpolation function. We have highlighted these corrections in Turquoise in the text.

Depth (km)	Pressure (Gpa)	Depth (km)	Pressure (Gpa)	Depth (km)	Pressure (Gpa)	Depth (km)	Pressure (Gpa)	Depth (km)	Pressure (Gpa)
0.00	0.00	150.00	4.78	600.00	21.04	1271.00	51.17	2471.00	111.82
3.00	0.03	185.00	5.95	600.00	21.04	1371.00	55.90	2571.00	117.35
3.00	0.03	220.00	7.11	635.00	22.44	1471.00	60.68	2671.00	122.97
15.00	0.34	220.00	7.11	670.00	23.83	1571.00	65.52	2741.00	126.97
15.00	0.34	265.00	8.65	670.00	23.83	1671.00	70.41	2741.00	126.97
24.40	0.60	310.00	10.20	721.00	26.08	1771.00	75.36	2771.00	128.71
24.40	0.60	355.00	11.77	771.00	28.29	1871.00	80.37	2871.00	134.56
40.00	1.12	400.00	13.35	771.00	28.29	1971.00	85.43	2891.00	135.75
60.00	1.79	400.00	13.35	871.00	32.76	2071.00	90.56	2891.00	135.75
80.00	2.45	450.00	15.23	971.00	37.29	2171.00	95.76	2971.00	144.19
80.00	2.45	500.00	17.13	1071.00	41.86	2271.00	101.04	3071.00	154.70
115.00	3.62	550.00	19.07	1171.00	46.49	2371.00	106.39		

Table 1 Corresponding data of pressure and depth from Preliminary Reference Earth Model

2. The conversion of $\text{CaC}_2\text{O}_5\text{-I42d}$ to $\text{CaC}_2\text{O}_5\text{-Fdd2}$ results in an increase in seismic wave velocity, yet the author does not discuss the implications of this change on mantle structure or composition. This aspect requires further exploration.

A: We think this is an excellent suggestion. Here are the two main points to answer your question:

① In our research, we found that the multiphase state of CaC_2O_5 may be generated under different pressure conditions in the lower mantle due to the discovery of CaCO_3 diamond inclusions and liquid CO_2 by previous researchers^{45,46}. This suggests that $\text{CaC}_2\text{O}_5\text{-Fdd2}$ may also be generated by the reaction of CaCO_3 and CO_2 . (at line 291-295)

② Additionally, $\text{CaC}_2\text{O}_5\text{-I42d}$ can exist stably under lower mantle pressure conditions (as shown in Fig. 1 and König et al.'s experimental data⁴) and has a wave velocity that is very similar to that of Cubic Ca-Pv ³¹. Therefore, we believe that the transformation from $\text{CaC}_2\text{O}_5\text{-I42d}$ to $\text{CaC}_2\text{O}_5\text{-Fdd2}$ may be very rare or very slow.

③ we have reviewed the discussion and conclusion sections of our article and systematically re-summarized the possible effects of CaC₂O₅-minerals under high pressure on mantle structure (line 293-305).

3. *The benchmark calculation should be included in the method section.*

A: Based on your feedback, we have moved The benchmark calculation to the Methods section of the article (line 347).

4. *Several corrections are recommended, such as modifying table and figure references, addressing typos, and ensuring proper formatting of journal names in the references.*

Line 68: Table 2 should be Table 1;

Line 74.75: reference 4.5 already cited in line 73 and not needed again

Line 78. “two pressure” should be “two pressures”

Line 116. “polymorphs is is constant” should be “polymorphs is constant”

Line 123. “ δ -AlOOH₂₀ and” should be “ δ -AlOOH₂₀, and”

Line 240: PREM model should be the PREM model

Line 259. “at the depth” should be “at a depth”

Line 283. delete “and”

Line 327: densitiy should be density

Figure 9: Ca₂CO₅ should be CaC₂O₅

Incorrect way of reference listed of References 2

A: We were really sorry for our careless mistakes. Thank you for your reminder. We have thoroughly polished and corrected the language and formatting of our manuscript using **Research Square AJE Services** to ensure the correctness of the language and formatting of our article. In our resubmitted manuscript, all changes have been highlighted in red. Thanks for your correction.

REVIEWERS' COMMENTS

Reviewer #3 (Remarks to the Author):

The authors have overall addressed my concerns. I recommend publishing this work as it is.